# The Role of CRABS CLAW Transcription Factor in Floral Organ Development in Plants

**DOI:** 10.3390/ijms26199377

**Published:** 2025-09-25

**Authors:** Piotr Szymczyk, Jadwiga Nowak, Małgorzata Majewska

**Affiliations:** 1Department of Biology and Pharmaceutical Botany, Medical University of Łódź, Muszyńskiego 1, 90-151 Łódz, Poland; 2Department of Pharmacology and Therapeutics, School of Biomedical Sciences, College of Health Sciences, Makerere University, Kampala P.O. Box 7062, Uganda; jagodanowak@hotmail.com; 3Department of Oncobiology and Epigenetics, Faculty of Biology and Environmental Protection, University of Lodz, Pomorska 141/143, 90-236 Łódź, Poland; malgorzata.majewska@biol.uni.lodz.pl

**Keywords:** CRABS CLAW, gynoecium, floral meristem, gene and protein interactions

## Abstract

CRABS CLAW (CRC) is a member of the plant-specific YABBY transcription factor family, defined by the presence of a C2C2 zinc-finger domain and a C-terminal YABBY domain. CRC is essential for proper floral development, functioning in the termination of the floral meristem, maintenance of adaxial–abaxial polarity within the gynoecium, and regulation of nectary and leaf morphogenesis. CRC orchestrates its diverse regulatory functions through interaction networks comprising other transcription factors and plant developmental regulators, including chromatin-modifying enzymes and proteins involved in auxin biosynthesis, transport, and signaling. The roles of genes and proteins interacting with *CRC* or CRC have been characterized in several model plant species, and the number of identified *CRC*/CRC-associated interactions continues to expand, revealing both species-specific and conserved functional roles across angiosperms. Many functions of *CRC* and its interacting partners have been elucidated through the analysis of anatomical and physiological phenotypes associated with specific gene mutations. The functional roles of *CRC* in plant development appear to have been acquired progressively through evolutionary diversification. These evolutionary changes have been associated with the relative conservation of *CRC* gene copy number and a predominant role of mutations occurring in non-coding regulatory regions. These properties are attributed to the relatively limited number of genes comprising the *CRC* regulatory network and the capacity to induce dosage-dependent effects via the emergence of novel proteins with overlapping or analogous functions. The identification and functional characterization of CRC transcription factors across diverse plant species has advanced rapidly in recent years, yet a comprehensive synthesis of these findings has not been presented in a dedicated article. Therefore, this study reviews the current knowledge on CRC transcription factors, with a focus on their identification, expression patterns, and functional roles in plant development.

## 1. Introduction

The *CRC* gene belongs to the YABBY transcription factor family and plays a pivotal role in meristem termination, carpel development, and nectary formation in plants [1,2,3,4,5,6]. Its ortholog in the monocot *Oryza sativa*, known as *DROOPING LEAF* (*DL*), regulates not only carpel identity but also the development of the leaf midrib [7,8,9]. Given the critical role of *CRC* in carpel development and floral meristem (FM) termination, its expression is subject to strict regulation, primarily mediated by other transcription factors [10,11,12].

The termination of plant meristem activity requires repression of the *WUSCHEL* (*WUS*) gene, which is achieved through two feedback loop systems [13,14]. In one of these loops, *WUS* is initially repressed either directly by AGAMOUS (AG) or indirectly via the AG-dependent activation of *KNUCKLES* (*KNU*) during early stages of floral development [13,14]. This repression is subsequently enhanced by the action of additional regulatory proteins, including CRC and TORNADO2 (TRN2), which contribute to the maintenance of auxin maxima necessary for proper meristem termination and gynoecium development [15,16,17,18]. Simultaneously, auxin represses cytokinin biosynthesis and signaling, a regulatory interaction essential for proper FM determinacy [18].

CRC and AG regulate the expression of *YUCCA4* (*YUC4*), a gene involved in the two-step conversion of tryptophan to indole-3-acetic acid (IAA), a key process in auxin biosynthesis [15,16]. Elevated local auxin concentrations resulting from *YUC4* activation promote cell wall loosening and cell proliferation, particularly within the medial domain of the developing gynoecium [12,13,14,15].

Analysis of phylogenetic studies on *CRC* genes suggests their numerical stability across angiosperms [19]. This feature is relatively uncommon among evolutionarily younger genes and may be attributed to the compact size of the gene regulatory network involved in carpel development [20,21]. The introduction of novel genes with overlapping functions may disrupt the delicate equilibrium, of this regulatory network, potentially causing dosage imbalances—particularly detrimental in processes such as the termination of reproductive meristems [20].

Ancestral *CRC* genes likely contributed to the polarity of the carpel—an organ unique to angiosperms—through subfunctionalization of the broader role of *YABBY* genes in establishing lateral organ polarity [1,22,23,24]. During angiosperm evolution, *CRC* expression and function underwent significant diversification, including the acquisition of roles in specifying carpel identity, the loss of polarity-regulating function in grass carpels, and the evolution of novel functions such as midrib development in grass leaves and nectary formation in core eudicots [1,7,8,9,11,24,25].

This review aims to synthesize current knowledge on the role of the CRC transcription factor in FM termination and the development of carpels and nectaries. Particular emphasis is placed on the identification and characterization of CRC-associated gene and protein partners, highlighting their contributions to the regulatory networks that coordinate plant growth, organ specification, and reproductive development.

## 2. Structural Features of the CRC Protein

The *Arabidopsis thaliana* CRC transcription factor belongs to the YABBY family and exhibits characteristic structural elements, including a C2C2-type zinc finger motif spanning amino acid residues 26–53, which confers DNA, RNA, and protein-binding capabilities [4,26] (Figure 1). The YABBY family is plant-specific and classified within the fourth superclass of all-α-helical DNA-binding domains, namely the High-mobility Group (HMG) domain factors. This class is conserved between plants and mammals [27]. The *A. thaliana* CRC protein (GenBank: KAL9311171.1) consists of 181 amino acids, with an intermediate segment located between residues 54–108 that likely functions as a flexible linker (Figure 1). This region contains a serine- and proline-rich domain at its center—a typical feature of transcription factor activation domains [28,29].

The YABBY domain of CRC is located between amino acid residues 109–155 and shares structural similarity with helix–loop–helix motifs found in HMG-box-containing proteins and plays a critical role in DNA binding [4,26] (Figure 1). However, in CRC, only the first two of the three helices characteristic of *A. thaliana* HMG1-like proteins are present [4,26,30].

Proteins exhibiting this domain organization were classified in the late 1990s as members of a newly defined YABBY transcription factor family. The family was named after the Australian freshwater crayfish “yabby”, in reference to its founding member, CRABS CLAW (CRC) [4,31,32].

Subcellular localization studies using *CRC* deletion mutants suggest the presence of a putative nuclear localization signal (NLS) within the YABBY domain, specifically spanning amino acid residues 110–117 (Lys-Pro-Pro-Lys-Glu-Lys-Lys-Gln-Arg; KPPEKKQR) (Figure 1). Despite the basic residue composition characteristic of classical NLSs, site-directed mutagenesis of this sequence does not abolish nuclear import, indicating that the NLS may exhibit non-canonical features [26,33]. This behavior is consistent with NLS motifs that interact with the minor binding site of importin α, as described for other atypical nuclear import signals [33,34,35].

Results from P-BLAST analysis using the *A. thaliana* amino acid sequence, followed by multiple sequence alignment, enabled the identification of highly conserved and less conserved amino acid positions based on the relative entropy threshold of each residue [36]. The data indicate that the strongest conservation occurs within the C2C2-type zinc finger motif, the N-terminal half of the Ser/Pro-rich domain, the NLS, and the N-terminal half of Helix 1 (Appendix A).

## 3. CRC in Floral Meristem Termination

Meristematic cells of the FM originate from the shoot apical meristem (SAM), which remains active throughout the plant’s life cycle and maintains a population of stem cells in its apex, referred to as the central zone (CZ) [37,38]. Cells in the CZ undergo slow divisions and are subsequently displaced laterally into the peripheral zone, developing further into lateral organs such as flowers or leaves [37,38].

In *A. thaliana*, the timing of FM termination is primarily regulated by AG, which controls the expression of *WUS*, *KNU*, and *CRC* through different feedback loop systems [39,40,41,42,43,44].

The first regulatory feedback loop involves WUS, which activates the expression of *CLAVATA3* (*CLV3*) in the CZ [43,44,45]. The activation of *CLV3* transcription requires the physical interaction between WUS and members of the HAIRY MERISTEM (HAM) protein family [46]. Additionally, WUS binds to the *CLV3* promoter as a heterodimer in association with SHOOT MERISTEMLESS (STM) [47]. In turn, CLV3 is perceived by receptor complexes composed of CLAVATA1 (CLV1), CLAVATA2 (CLV2), RECEPTOR-LIKE PROTEIN KINASE 2 (RPK2), CLAVATA3 INSENSITIVE RECEPTOR KINASES (CIKs), CORYNE (CRN), and BARELY ANY MERISTEMS (BAMs) [48]. These complexes mediate the repression of *WUS* expression, thereby regulating stem cell maintenance in the shoot apical meristem [37,48,49,50] (Figure 2A).

A related but independent feedback loop involves the CLE40 peptide, which is closely related to CLV3 and promotes *WUS* expression through the CLV1-family receptor BAM1 [51]. In turn, *CLE40* expression is repressed in a *WUS*-dependent manner, establishing a regulatory circuit that contributes to meristem homeostasis [51] (Figure 2B).

AG plays a central role in the second feedback loop that regulates the termination of *WUS* expression and is overlaid on the CLV-WUS pathway [44,52]. Mutations in *AG* disrupt this regulation, leading to the failure of *WUS* repression and resulting in the formation of supernumerary floral whorls from the floral center. This produces a characteristic ‘flower-within-a-flower’ phenotype, also referred to as the Russian doll phenotype [4,53].

The mechanisms by which AG regulates *WUS* expression are dependent on the developmental stage of the floral bud (Figure 3A). During floral stage 3, *WUS* promotes the expression of the C-class gene *AG* and *LEAFY* (*LFY*) [54]. The floral identity protein LEAFY (LFY) cooperates with WUS to activate *AG* in the center of flowers [54]. Subsequently, AG represses *WUS* by recruiting the TERMINAL FLOWER 2 (TFL2) protein, a component of the Polycomb Repressive Complex 1 (PRC1) [13,55]. In a later developmental stage (stage 6), AG directly enhances the expression of the C2H2-type zinc finger protein KNU, initiating a two-step repression mechanism of *WUS* expression [14,56,57]. Initially, *WUS* expression is repressed through the association of KNU with a histone deacetylase complex [56]. This initial repression is subsequently stabilized by KNU-mediated recruitment of the Polycomb Repressive Complex 2 (PRC2), which catalyzes the trimethylation of lysine 27 on histone H3 (H3K27me3), a hallmark of stable gene silencing (Figure 3A) [14,58]. The influence of KNU, EPFL and GPA1 on the WUS-CLV3 regulatory route and *CRC* expression was presented in Figure 3B [3,13,43,56,59,60,61,62,63,64,65,66].

Representative genes that are either directly regulated by AG or integrated within the AG regulatory network are listed in Appendix A [2,42,67,68,69,70,71,72].

It is plausible that CRC alone is not sufficient to modulate *WUS* activity, as suggested by analyses of *CRC* mutants across multiple species, including *A. thaliana*, pomegranate, and rice, which exhibit no overt FM defects [7,25,73].

In *A. thaliana*, CRC appears to play an ancillary role to AG in FM development [11,25,68,74]. The heterotetramer formed by SEP3 and the MADS protein AGAMOUS is necessary to activate two target genes, *KNU* and *CRC*, which are required for meristem determinacy [75,76]. However, more direct and independent roles of CRC have been observed in tomato, where it contributes to high temperature-induced meristem termination [3,39]. In tomato, *Sl*CRCa and *Sl*CRCb directly bind to the *SlWUS* promoter, with *Sl*CRCb also interacting with the second intron of *SlWUS* [39]. Interestingly, *At*CRC does not bind to the *AtWUS* promoter (p*AtWUS*) or its genomic locus (g*AtWUS*), suggesting a species-specific regulatory mechanism [39].

Under heat shock conditions, a reduction in brassinosteroid concentrations within the floral meristem results in the persistent downregulation of *SlCRCa* [39]. This failure to terminate *SlWUS* expression ultimately leads to the development of malformed fruits [39]. Although previous studies have demonstrated the critical role of tomato AG1 (*TAG1*) in activating *SlKNU* and *SlIMA*, leading to repression of *WUS* expression, overexpression of *TAG1* alone is insufficient to activate the promoter activity of *SlKNU* and *SlIMA*. Therefore, in tomato, *SlCRCa* is necessary for *TAG1*-mediated activation of *SlIMA* and *SlKNU* [56].

Dual-luciferase assays suggest that this missing transcriptional activation may be provided by *Sl*CRCa, but not by *Sl*CRCb, indicating functional divergence between the two CRC homologs in tomato [26,56]. Moreover, *Sl*CRCa directly binds to the *SlWUS* locus, where it recruits the *Sl*IMA complex, leading to complete repression of *SlWUS* expression during the establishment of FM determinacy [3]. More specifically, *Sl*CRCa and *Sl*CRCb proteins can interact with one another, and each can also interact with *Sl*KNU, *Sl*IMA, and *Sl*HDA1. Notably, only *Sl*CRCb, and not *Sl*CRCa, interacts with *Sl*TPL1 [3].

## 4. CRC in Auxin/Cytokinin Crosstalk

Further studies have demonstrated that the pivotal role of AG in FM termination and gynoecium development is mediated through its regulation of the crosstalk between cytokinin, miRNA and auxin signaling pathways [16,77,78,79,80]. Similarly, findings by Yamaguchi et al. (2017,2018) highlight the essential function of CRC in FM termination and gynoecium formation via the fine-tuning of auxin homeostasis [12,13,15,16,81]. Moreover, the transcriptional repressor KNU integrates and modulates the activities of auxin and cytokinin, thus securing timed FM termination [82]. The roles of CRC and AG in regulating auxin and cytokinin activity are presented in Figure 4 [80,83,84,85,86,87,88,89,90,91,92,93,94].

Although AG represses *WUS* directly or via *KNU* during floral stages 3–5, this repression is further reinforced at stage 6 by additional regulatory factors, including CRC and TRN2. CRC represses *TRN2*, leading to the formation of local auxin maxima (Figure 4) [15,16]. Concurrently, AG and auxin synergistically activate *ARF3*, which in turn downregulates the expression of *ISOPENTENYL TRANSFERASES (IPTs)*, *LONELY GUY (LOG)*, and *ARABIDOPSIS HISTIDINE KINASE 4 (AHK4*), thereby attenuating cytokinin signaling activity (Figure 4) [77]. The increase in auxin activity, combined with the decrease in cytokinin motion, results in the indirect inhibition of *WUS* expression, or WUS destabilization, leading to floral meristem termination (Figure 5) [15,16,77,81,95,96,97,98,99,100,101].

The pivotal role of auxin in the regulatory functions of CRC and AG during FM termination has been elucidated through transcriptomic analyses of relevant mutants, including *crc-1*, *knu-1 versus knu-1*, and *ag-12* versus wild-type plants. These comparisons led to the identification of a core set of 53 genes that are co-regulated by both CRC and AG [15]. Among the 53 genes co-regulated by AG and CRC, a subset of nine genes were identified as direct AG targets based on publicly available AG ChIP-seq datasets [15]. Among these, *YUC4* is an auxin-related target of *CRC*, acting downstream and synergistically with *TRN2* to promote FM termination through the indirect repression of *WUS* (Figure 4) [12,15].

YUC4, in conjunction with TRYPTOPHAN AMINOTRANSFERASE of ARABIDOPSIS (TAA) family enzymes, catalyzes a two-step conversion of tryptophan to indole-3-acetic acid (IAA), representing a critical phase in the auxin biosynthesis pathway [17,99,102,103].

Putative synergistic and positive regulation of *YUC4* by *CRC* and *AG* was supported by qRT-PCR analysis in *crc-1* and *ag-1-/+ crc-1* double mutants. Notably, YUC4 reporter (GUS) expression in the abaxial carpels at floral stage 6 was markedly reduced in the *ag-1-/+ crc-1* double mutants compared to the *crc-1* single mutant [15]. Induced expression of *AG* and *CRC*, either individually or in combination, supported a synergistic role in the activation of *YUC4* transcription [15].

The regulation of *YUC4* by CRC involves its binding to YABBY-binding motifs (GA[A/G]AGAAA) located within conserved regulatory modules 1–4 (CRMs 1–4) of the *YUC4* promoter [12,104].

Upregulated expression of *YUC4* enhances local auxin accumulation, leading to cell wall loosening and increased cell proliferation, potentially facilitated by heightened cell division activity in the medial domain of the developing gynoecium [16,105]. The dynamic distribution of the plant morphogen auxin is pivotal for the bilateral-to-radial symmetry transition, which is required for proper formation of the apical style in the *Arabidopsis* gynoecium [106].

*At*CRC and *At*AP1 may participate in the regulation of very long-chain fatty acid (VLCFA) biosynthesis. In *Arabidopsis*, VLCFAs regulate polar auxin transport and lateral root organogenesis (Figure 4) [99,100].

## 5. CRC Interaction Networks

Mutations in the *CRC* gene in *A. thaliana* lead to a distinct floral phenotype, characterized by a wider and shorter gynoecium, unfused carpels at the apex, reduced ovule number, and a complete absence of nectary development at all stages of floral maturation [4]. The gene was mapped between *CLV2* and *CLV1* on chromosome 1, subsequently positionally cloned, and effectively used to complement the *crc* mutation, thereby suppressing early radial expansion of the gynoecium and promoting its elongation at later developmental stages [4,25,43].

The inhibition of radial growth in the developing gynoecium by CRC begins in the lateral regions of the gynoecial primordium and is later reinforced within the epidermal layer. Although CRC also functions in the early stages of nectary development, its regulatory interactions differ between the gynoecium and nectary tissues. In the gynoecium, *CRC* expression is negatively regulated by A-class genes, including *APETALA2* (*AP2*) and *LEUNIG* (*LUG*) [4]. The same holds true for the B-class genes in *Arabidopsis*, *PISTILLATA* (PI) and *APETALA3* (AP3), which negatively regulate *CRC* expression in the third whorl [4]. Since mutations in the C-class gene *AGAMOUS* (AG) disrupt carpel development, the effect of *AG* loss-of-function was assessed in the first whorl. These studies showed that *CRC* expression persists in the absence of *AG*, but is significantly reduced and displays altered spatial distribution. Notably, *CRC* expression in nectaries remains unaffected by mutations in A-, B-, or C-class genes [4]. Genes and proteins associated with CRC regulation and the biological outcomes of their interactions are presented in Appendix A [4,12,17,39,62,73,77,83,100,107,108,109,110,111,112,113,114].

In *A. thaliana*, the gynoecium is composed of two congenitally fused carpels, forming a single-chambered ovary (Figure 6) [115,116,117].

Abaxial–adaxial polarity within the gynoecium is evident from distinct epidermal cell morphologies on the abaxial (outer) and adaxial (inner) surfaces of the ovary valves, which contribute to proper organ development (Figure 6) [107].

A novel mutant phenotype in *Arabidopsis* carpels, characterized by the duplication of adaxial tissues—such as placentae and ovules—at abaxial positions, in addition to their normal localization on the adaxial (internal) side, was observed in double mutants where the function of *CRC* and either *GYMNOS* (*GYM*) or *KANADI* (*KAN*) is compromised [106,107]. The ectopic development of adaxial tissues is normally suppressed by *CRC* and *KAN*, which act through independent pathways and provide positional information to specify abaxial cell fate in the developing carpel (Appendix A). The patterning events in the developing gynoecium depend on the activity of *SEUSS* and *AINTEGUMENTA* genes [108]. Phenotypic analyses of mutants for both genes suggest their synergistic interaction in the development of ovules and the medial domain of the gynoecium. Identified downstream targets of *SEUSS* and *AINTEGUMENTA* include *PHABULOSA* (*PHB*), *REVOLUTA* (*REV*), and *CRC* genes (Appendix A) [107]. Both *SEU* and *ANT* promote *CRC* expression in the internal domains of the gynoecial ovary. However, in the apical region of the gynoecium, *SEU* and *ANT* act to repress *CRC* expression within the medial ridge, which correlates with an increased incidence of carpel separation spanning approximately 90% of the apical–basal axis of the gynoecium (Appendix A) [107,108].

JAIBA *(*HAT1*),* a member of the class II HD-ZIP transcription factor subfamily, is implicated in both male and female reproductive development. In *jab* homozygous mutants, defects include a reduced number of mature pollen grains and ovules lacking a functional embryo sac [108,109,118]. Double *jaiba crc-1* mutants exhibit severe defects in FM determinacy and development of the gynoecium medial domain, indicating a functional relationship with *CRC* (Appendix A) [108,109]. The *jab crc-1* double mutant produced flowers with up to six sepals, up to six petals, and between five and seven stamens (Appendix A) [108,109]. Additionally, the gynoecia of these flowers were composed of two to four carpels, which at later developmental stages transformed into carpeloid structures within the fruit.

Gynoecia lacking functional *STYLISH1 (STY1)* expression due to a transposon insertion exhibit aberrant style morphology. These defects are further enhanced in *sty1 sty2* double mutants, which show a marked reduction in stylar and stigmatic tissues as well as decreased proliferation of stylar xylem [110]. *CRC* is proposed to interact with *STY1*, as *sty1-1 crc-1* double mutants display a more severe reduction in stylar, stigmatic, septal, and medial xylem tissues compared to either single mutant (Appendix A) [110]. STY1 functions as a transcriptional activator of the gene encoding the flavin-containing monooxygenase THREAD/YUCCA4, a key enzyme in the auxin biosynthesis pathway, ultimately resulting in altered auxin homeostasis [111,119].

The functional activity of transcription factors is frequently mediated not by monomeric forms but through the formation of homo- or heterodimers, which may assemble in the cytoplasm prior to nuclear translocation or upon binding to closely spaced cis-regulatory motifs within promoter regions [52,53,75,76,120,121]. Dimerization is a well-documented feature among members of the YABBY transcription factor family [26,32,44,48]. In the case of CRC, bimolecular fluorescence complementation (BiFC) assays suggest that dimer formation is dependent on the presence of the YABBY domain [26,44].

However, these CRC-CRC or CRC-YABBY protein interactions appear to be relatively weak and could not be confirmed using yeast two-hybrid (Y2H) assays [26,44]. Interestingly, Y2H analyses have indicated a weak interaction between the transactivation domain of CRC and the YABBY domain of the INNER NO OUTER (INO) protein, suggesting possible functional crosstalk between these two transcription factors (Appendix A). CRC-interacting partners may exert either activating or repressive effects on CRC-mediated transcriptional regulation [26,44]. Further evidence from Y2H and BiFC assays in *Punica granatum* (pomegranate) demonstrated that both CRC and INO interact with a common partner, *Pg*BEL1, implicated in the maintenance of ovule identity (Appendix A) [73].

The cooperative function of the YABBY domain and the zinc finger motif in DNA binding is supported by Y1H assays. However, Y1H screens using isolated or mutated CRC domains revealed that the YABBY domain plays a predominant role in DNA binding, while the zinc finger contributes a weaker, auxiliary interaction with the target DNA sequence [26].

In silico analysis of the regulatory mechanisms governing *CRC* expression suggests a predominant role for transcription factors, as indicated by the presence of only two CHH-context DNA methylation sites located approximately 0.3 to 3 kb upstream of the transcription start site (TSS). Furthermore, the *CRC* genomic locus appears to be largely unaffected by microRNA-mediated regulation and exhibits only moderate epigenetic control. This is supported by the presence of the repressive histone mark H3K27me3, which spans most of the *CRC* locus, including the promoter region, and by H2AK121ub enrichment within the transcribed region [12].

To identify putative transcription factors interacting with the *CRC* promoter, yeast Y1H screening was performed, yielding 140 candidate proteins. Of these, only 48 were annotated in the PlantPAN3 database and exhibited predicted DNA-binding motifs matching sequences within the *CRC* promoter region (Appendix A) [12].

Functional partners of CRC predicted by the STRING database are presented in Figure 7 [122]. Most of them are related to the known CRC functions in floral organ development, FM determinacy, or fatty acid homeostasis [3,4,100,102]. However, novel functionalities associated with plant defense or myo-inositol hexakisphosphate and phytic acid metabolism were observed (Figure 7). Detailed information concerning CRC functional partners is provided in Appendix A.

## 6. Evolutionary Diversification of CRC Gene

The origin of *CRC* coincided with the emergence of angiosperms, as *CRC* plays a pivotal role in the development of the carpel—an organ unique to flowering plants [20,123]. Although closely related *CRC* paralogs are present in gymnosperm genomes, it remains unclear how *CRC* was integrated into gene regulatory networks to control carpel-specific functions during the evolution of angiosperms [20,124].

Phylogenetic analyses indicate that *CRC* genes involved in FM termination, carpel organ identity, and abaxial-adaxial polarity in *A. thaliana* have remained numerically stable and have not undergone significant expansion across plant lineages [20,124].

The *CRC* gene is maintained as a single-copy gene in most angiosperms, which is atypical for evolutionarily recent genes [20]. It is hypothesized that the evolution of a relatively small and tightly regulated carpel gene regulatory network (GRN)—comprising genes such as *ALCATRAZ* (*ALC*), *CRC*, *HALF FILLED* (*HAF*), *HECATE* (*HEC*), *INDEHISCENT* (*IND*), *NGATHA* (*NGA*), and *SPATULA* (*SPT*)—does not tolerate dosage imbalances or the expansion of gene families encoding proteins with redundant functions [20].

The termination of reproductive meristems appears particularly sensitive to dosage imbalance, as demonstrated by the effects of increased *CRC* expression. Artificial elevation of *CRC* levels can lead to premature termination of reproductive meristems—an outcome that is strongly selected against in natural populations [1,125].

Although *CRC* is typically maintained as a single-copy gene in most plant species, an exception is observed in the *Solanaceae* family, where it occurs as paralogous pairs—*CRCa* and *CRCb* in *Solanum lycopersicum*, and *CRC1* and *CRC2* in *Petunia hybrida* [126]. These paralogs are believed to have arisen from a large segmental duplication event that likely occurred in a common ancestor of the *Solanaceae* [126].

The *CRC* ortholog in the monocot *Oryza sativa*, known as *DL*, exhibits differences in both spatial expression and function compared to *CRC* [7,8,9]. In *A. thaliana*, *CRC* is expressed in the abaxial region of carpel primordia and in floral nectaries, where it plays a key role in regulating carpel morphology and nectary development [1,3,5,6,7,12]. In *Oryza sativa*, *DL* is expressed throughout the entire carpel primordium and in the central undifferentiated cells of developing leaves, where it regulates both carpel identity and midrib development [7].

Analysis of *CRC* orthologs in two *Fabaceae* species, *Pisum sativum* and *Medicago truncatula*, revealed not only an atypical absence of abaxial expression in the carpel but also an unusual expression pattern associated with the medial vein of the ovary. This suggests a potential role in vascular development—previously thought to be specific to *DL* in monocotyledons [127].

The temporal and spatial expression patterns of *DL* orthologs in three grass species—maize, wheat, and sorghum—closely mirror those of *O. sativa DL* during both floral and leaf development. These findings suggest a high degree of functional conservation of *DL*-related genes within the *Poaceae* (grass) family [128].

Analysis of the spatial expression patterns of *CRC/DL* orthologs in eudicots and basal eudicots—such as *Petunia hybrida*, *Gossypium hirsutum*, *Eschscholzia californica* (basal eudicot), and *Aquilegia formosa* (basal eudicot)—reveals that the abaxial specificity and absence of expression in leaves, characteristic of *A. thaliana CRC*, are highly conserved among these species. This conservation suggests a stable and ancestral regulatory role for *CRC*/*DL* orthologs in gynoecial development across eudicot lineages [1,11].

The ancestral role of *CRC* in carpel development was confirmed in the basal angiosperm *Amborella trichopoda*, whose *CRC* orthologue partially complemented the *crc-1* mutant phenotype in carpels but not in nectaries [129]. This partial complementation suggests that novel functions of *CRC* orthologues, such as nectary development, may be mediated by evolutionary changes in non-coding regulatory sequences [130].

*CRC* also regulates the formation of nectaries, which are present not only in the floral nectaries of *Arabidopsis* but also in the extrafloral nectaries of *Gossypium hirsutum* and *Capparis flexuosa* [4,131]. However, in basal eudicots such as *Aquilegia formosa*, the *CRC* orthologue is not expressed in the nectary, suggesting that *CRC* orthologues were recruited for nectary development at the base of the core eudicots lineage [130,131,132].

Studies of *CRC*/*DL* expression across major angiosperm clades demonstrate the stepwise acquisition of novel functions without gene duplication, exemplifying the role of regulatory elements—such as promoters and enhancers—in broadening gene function (Figure 8) [1,24,130,131]. Alternatively, the novel function of *CRC* in nectary development may have arisen through one or more changes affecting the composition of downstream target genes, the availability or identity of protein interaction partners, and/or the spatiotemporal expression of *CRC* in nectary tissues [130].

Analysis of the functional diversification of *DL* in *Asparagus asparagoides* revealed that the evolution of *CRC*/*DL* genes from their ancestral role in specifying abaxial cell fate during carpel development to acquiring expression in leaves occurred prior to the divergence of the order *Asparagales*. However, traits such as expression throughout the entire carpel primordium, in the central region of leaves, and involvement in carpel organ identity were gained after the divergence of *Asparagales* (Figure 8) [24].

In the orchid *Phalaenopsis equestris*, two *DL*/*CRC* paralogs, *PeDL1* and *PeDL2*, were identified as products of a whole-genome duplication (WGD) event in the last common ancestor of orchids. Both genes are expressed in the FM and carpel tissue, supporting the ancestral role of *DL*/*CRC* genes in FM determinacy and carpel specification [23].

Moreover, expression of both *PeDL* genes was observed in the placenta and ovule primordia during early stages of ovary development in *Phalaenopsis equestris* [133]. Similarly, in California poppy (*Eschscholzia californica*), the *CRC* ortholog *EcCRC* has also been co-opted for additional roles in ovule initiation [1].

## 7. Polymorphisms and Natural Variation of the CRC Gene

T-DNA and ethyl methanesulfonate (EMS) mutagenesis screens performed in the *crc-1* background led to the identification of novel genetic modifiers of the *crc-1* phenotype. These were cloned and characterized as *rebelote-1 (rbl-1,* meaning “once again” in French), *squint-4 (sqn-4),* and *ultrapetala1-4 (ult1-4).* In all three resulting double mutants, the FM remains indeterminate, with ectopic stamens and carpels arising from the central region of the FM, between the primary carpels, and along an elongated floral axis [125].

*RBL, SQN,* and *ULT1* function redundantly in the regulation of FM termination, likely through modulation of *WUS* activity [125].

Moreover, the *rbl-1*, *sqn-4,* and *ult1-4* mutations enhance the ag-4 phenotype, suggesting that *RBL*, *SQN,* and *ULT1* play partially redundant roles in supporting *AG* function [33]. Additionally, genetic crosses with the *ag-6* mutant indicate that *RBL, SQN,* and, to a lesser extent, *ULT1* contribute to FM determinacy via an alternative pathway, potentially involving *SUPERMAN (SUP),* as the phenotypes of *ag-6 rbl-1* and *ag-6 sqn-4* resemble those of the *ag-1 sup-1* double mutant [125,131].

Mutations in the tomato (*Solanum lycopersicum*) *CRCa* gene (*SlCRCa*) lead to the development of abnormal carpels that grow concentrically, one inside another, resulting in iterative fruit morphologies characteristic of the *fig* mutant phenotype [3]. In addition, affected fruits frequently develop secondary fruit structures that initiate internally and progressively expand to emerge on the fruit surface [3]. The molecular basis of the *fig* mutation is a 365-bp insertion of a putative transposable element within the intron between exons 4 and 5 of the *SlCRCa* gene [3]. The regulation of fruit size changes induced by *SlCRCa* may also involve *YABBY5a* and genes of the gibberellin biosynthesis pathway, both of which are generally downregulated in response to *SlCRCa* overexpression [132].

Although *SlWUS* expression is terminated from stage 6 onward in wild-type plants, *SlCRCa* and *SlCRCb* mutants maintain *SlWUS* expression beyond stage 6, resulting in defects in floral determinacy [3]. Similar to *A. thaliana*, the *SlWUS* locus in tomato is associated with a chromatin remodeling complex exhibiting histone deacetylase activity, as demonstrated by BiFC and co-immunoprecipitation assays.

Analysis of quantitative trait loci (QTL), in combination with map-based cloning, led to the identification of a nonsynonymous polymorphism (G to A) in the *CRC* gene of cucumber (*Cucumis sativus*, *CsCRC*) as the causal factor underlying variations in fruit size and shape. RT-PCR analyses revealed that *CsCRC* expression is predominantly confined to reproductive organs, including flowers and fruits, with no detectable expression in vegetative tissues such as stems and leaves. Among reproductive organs, the highest *CsCRC* expression was observed in the nectary, followed by female flower buds and young ovaries [133].

The presence of a rare *CsCRC^A* allele is associated with reduced fruit length in *Cucumis sativus*, whereas the *CsCRC^G* allele shows a positive correlation with increased fruit size. The causal variant is a nonsynonymous SNP (G-to-A) in *CsGy5G023910*, a homolog of *Arabidopsis CRC*, which leads to an amino acid substitution from arginine (R) to glutamine (Q) within the conserved YABBY DNA-binding domain [133]. The arginine-to-glutamine (R→Q) substitution within the YABBY DNA-binding domain of CsCRC markedly diminishes its binding affinity to [A/T]ATCAT[A/T] and [T/A]ATGAT[T/A] cis-regulatory motifs in the *CsARP1* promoter, thereby reducing *CsARP1* expression [133].

Putatively, the upregulated *CsARP1* transcription mediated by *Cs*CRC wild type enhances cell expansion in the pulp cells of cucumber fruit mesocarp, potentially via transmembrane electron transport and cell wall modification, as implied by the presence of a conserved cytochrome b561 domain and an additional DOMON domain in the encoded protein [133].

These findings support a critical link between *CRC* function and auxin-regulated pathways, consistent with earlier studies [15,16,81].

## 8. Outlook

Studies on the function of *CRC* in meristem termination and gynoecium or nectary development were initially conducted in the model plant *A. thaliana* [4,11,12,26,37,109,131,134]. The phenotypic consequences of *CRC* gene mutations, together with the availability of the *A. thaliana* genome sequence and transcriptomic datasets, provided a foundation for the identification of novel genes that are either co-expressed with *CRC* or modulate its role in meristem termination and gynoecium development [12,21,100,107,125,135].

Furthermore, Y2H and BiFC assays have yielded critical insights into proteins that interact with CRC or AG and directly contribute to the termination of *WUS* expression [17,26,73].

Beyond *A. thaliana*, *CRC* paralogs have been relatively well studied in tomato, providing insights into species- or developmental stage-specific variations in the composition of protein complexes involved in *WUS* repression [3,39,56]. However, studies investigating *CRC* function in other plant species remain relatively limited [1,10,24,111,136,137,138].

Consequently, future research should focus on the isolation of *CRC* orthologs and paralogs in a broader range of plant species. Generating transgenic lines with *CRC* overexpression, gene silencing, or targeted mutations would provide a basis for analyzing phenotypic and physiological alterations associated with *CRC* function.

These effects could be compared with observations from *A. thaliana*, rice and tomato to elucidate both conserved and novel roles of the *CRC* gene. In newly studied plant species, *CRC* gene mutations could be combined with chemical mutagenesis or transcriptomic analyses to identify previously uncharacterized genes that are co-expressed with *CRC* and modulate phenotypic or physiological traits associated with meristem termination and gynoecium development. In this way, the already known and novel functionalities of *CRC* could be revealed.

The application of Y1H assays or chromatin immunoprecipitation (ChIP) techniques could facilitate the identification of trans-acting factors regulating *CRC* gene expression in novel, often non-model, plant species. Similarly, Y2H, Y3H, and BiFC assays could be employed to identify components of protein complexes centered around CRC that are directly involved in executing its function. The functional relevance of these interactions could be validated by analyzing phenotypic alterations in plants harboring mutations in CRC-associated proteins.

Y2H screening often yields a substantial number of putative interacting proteins, frequently exceeding one hundred, many of which are false positives or are partially annotated in current databases [139,140,141,142,143,144,145]. Moreover, future progress in identifying cis-elements or trans-acting factors regulating *CRC*-dependent gene expression will rely upon advancements in the functionality and comprehensiveness of plant genomic and proteomic databases and improvements in Y1H methodology [140,141,142,145,146].

## Figures and Tables

**Figure 1 ijms-26-09377-f001:**
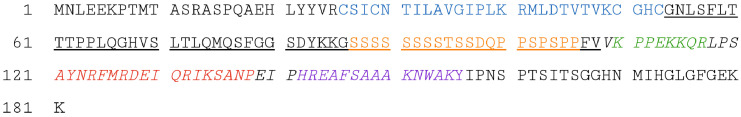
The domain organization of *A. thaliana* CRC protein (GenBank: KAL9311171.1). The following domains are presented and their span is marked: C2C2 zinc finger-like domain (26–53; blue); Flexible Linker (FL) (54–108; underlined); Ser/Pro-rich domain (87–106; orange); NLS (110–117; green), YABBY (109–155; italics), Helix 1 (121–138; red) and Helix 2 (142–156; purple). Some domains such as FL and Ser/Pro-rich domain, YABBY and Helix 1, YABBY and Helix 2 or NLS and YABBY are overlapping.

**Figure 2 ijms-26-09377-f002:**
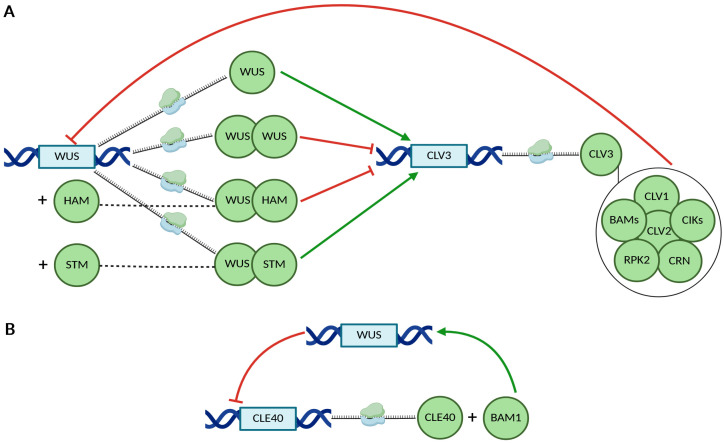
Basic regulatory mechanisms of WUS and CLV3 in a feedback loop. (**A**) The role of WUS monomers and homo- or heterodimers in regulating *CLV3* expression. (**B**) Reciprocal regulation of *WUS* and *CLE40* mediated via the BAM1 receptor. Green arrows indicate activation of gene expression, whereas red half-lines represent transcriptional repression. Protein names are shown within green, geometric circle shapes. Interacting proteins are shown as partially overlapping green circles or thin black lines. Protein complexes are presented inside larger circles. Gene names are shown within blue rectangles.

**Figure 3 ijms-26-09377-f003:**
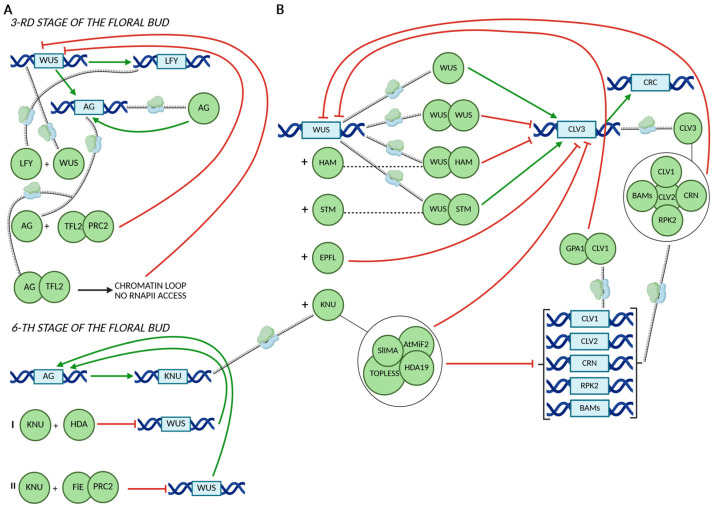
Mechanisms regulating WUS expression at the 3rd and 6th stages of floral bud development. At stage 3, AG plays a pivotal role either by recruiting PRC2 to the *WUS* promoter through TFL2, or by forming a chromatin loop with TFL2 that prevents RNAPII access. At stage 6, AG activates *KNU* expression, which, together with HAD and PRC2, represses *WUS* expression (**A**). The effects of KNU, EPFL, and GPA1 on the WUS–CLV3 regulatory circuit and *CRC* expression are shown in (**B**). Green arrows indicate gene expression activation, whereas red half-lines represent transcriptional repression. Protein names are displayed within green, geometric circle shapes. Interacting proteins are shown as partially overlapping green circles or thin gray lines. Protein complexes are presented inside larger circles. Gene names are shown within blue rectangles.

**Figure 4 ijms-26-09377-f004:**
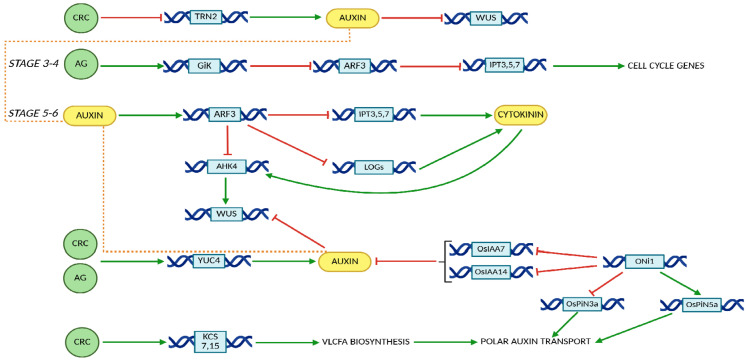
Roles of CRC and AG in regulating auxin and cytokinin activity. Green arrows indicate gene expression activation, whereas red half-lines represent transcriptional repression. Protein names are displayed within green, geometric circle shapes. VLCFA; Very Long Chain Fatty Acids. Orange dotted lines indicate the shared auxin pool, modulated by distinct transport and regulatory pathways. Gene names are shown within blue rectangles.

**Figure 5 ijms-26-09377-f005:**
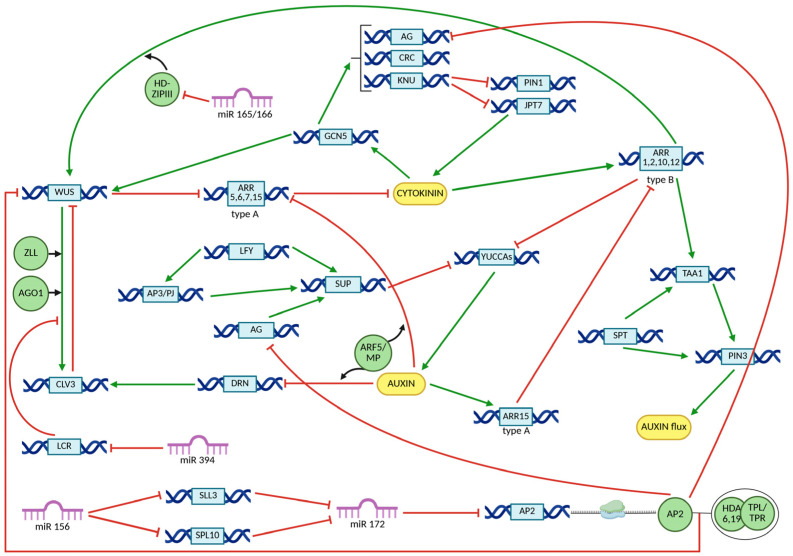
Influence of auxin, cytokinin, and miRNAs on *WUS* and *CLV3* gene expression. Green arrows indicate activation of gene expression, while red half-lines represent transcriptional repression. Protein names are displayed within green, geometric circle shapes. Interacting proteins are shown as partially overlapping green circles or thin gray lines. Protein complexes are presented inside larger elipses. Gene names are shown within blue rectangles.

**Figure 6 ijms-26-09377-f006:**
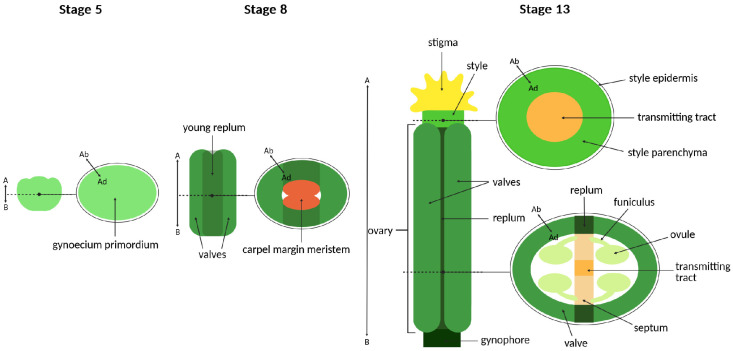
Structure and developmental stages of the gynoecium in *A. thaliana.* Schematic models and transverse sections at stages 5, 8 and 13. Dashed lines indicate the positions of the cross sections. A—apical; B—basal; Ad—adaxial; Ab—abaxial [116].

**Figure 7 ijms-26-09377-f007:**
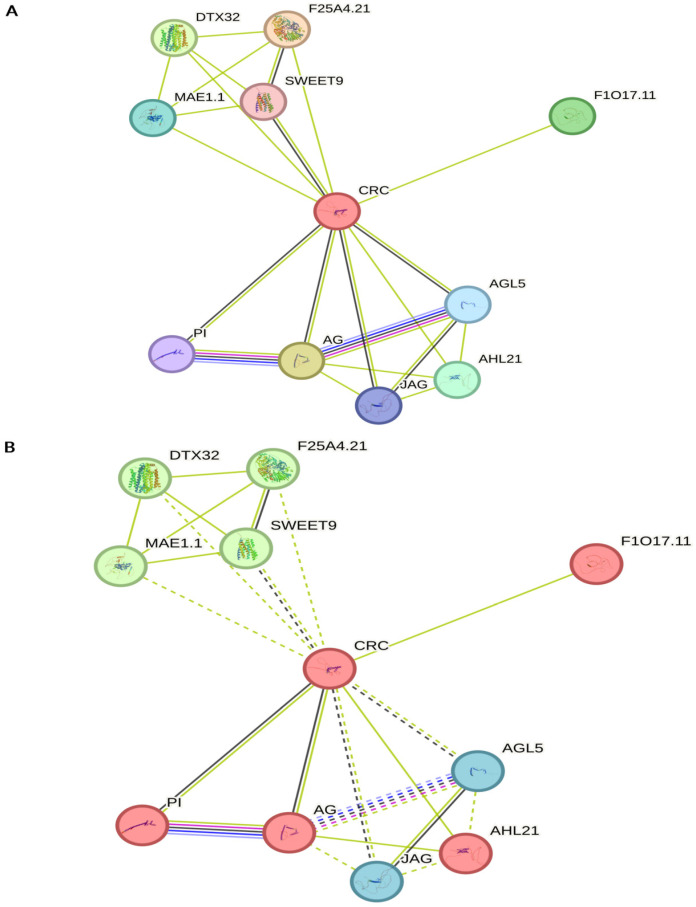
Functional partners of CRC predicted by the STRING database (**A**) and k-means clustering of obtained results (**B**) [122]. Among the three clusters, the largest (five members, marked in red) is associated with fatty acid homeostasis and floral meristem determinacy; AGAMOUS (AG), CRABS CLAW (CRC), PISTILLATA (PI), Homeobox-like protein F1O17.11, and AT-hook motif nuclear-localized protein 21 (AHL21). The second cluster comprises four proteins (marked in green), linked to nectar secretion, Myo-inositol hexakisphosphate and phytic acid metabolism as well as plant defence; DETOXIFICATION 32 (DTX32), Germin-like protein subfamily T member 3 (F25A4.21), Bidirectional sugar transporter SWEET9, and P-loop NTPase domain-containing protein LPA1 homolog 1 (MAE1.1). The third cluster (marked in blue) includes two proteins; Agamous-Like 5 (AGL5) and JAGGED (JAG), related to floral and lateral organ development. Basic properties of CRCs functional partners are presented in Appendix A. Solid and dotted lines represent different types of functional associations: blue (**—**) known interactions from curated databases, purple (**—**) experimentally validated interactions, green (**—**) predicted interactions based on gene neighborhood, red (**—**) predicted interactions inferred from gene fusions, dark blue (**—**) predicted interactions based or gene co-occurrence, grey (**—**) associations based on text mining, black (**—**) co-expression based associations, pale purple (**—**) interactions inferred from protein homology.

**Figure 8 ijms-26-09377-f008:**
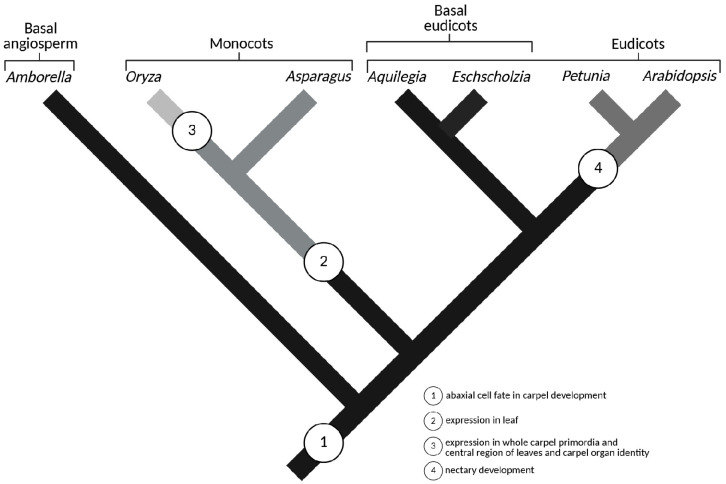
Evolutionary stages of CRC/DL subfamily genes across major angiosperm clades [24]. Line colors indicate the evolutionary state of orthologs: black lines represent the ancestral state (1), while grey lines in varying shades correspond to derived states (2–4).

## Data Availability

No new data were created or analyzed in this study. Data sharing is not applicable to this article.

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
