# Peer review of "The Role of CRABS CLAW Transcription Factor in Floral Organ Development in Plants"

_ijms, 2025, doi:10.3390/ijms26199377_

Round 1

Reviewer 1 Report

Comments and Suggestions for Authors

This is an ambitious and comprehensive review covering the CRABS CLAW (CRC) transcription factor across various areas, including structural biology, molecular interactions, developmental genetics, evolutionary history, and comparative plant biology. The literature coverage is extensive, referencing both classical and recent work (up to 2025). A significant strength of this review is the integration of molecular, phenotypic, and phylogenetic perspectives.

The article would serve as a valuable resource for researchers in plant developmental biology; however, several issues related to narrative clarity, structural organization, and conciseness should be addressed before publication.

The review provides impressive depth, consolidating structural, regulatory, and evolutionary aspects of CRC biology that will benefit both newcomers and experienced researchers. It incorporates recent literature up to 2025, including advancements such as heat stress-induced effects on CRC in tomato, naturally occurring CRC alleles in cucumber, and the identification of orchid paralogs, which ensures its scientific relevance. The work excels in cross-species comparisons, clearly distinguishing conserved functions from species-specific roles across monocots and eudicots. Additionally, it offers a detailed account of CRC’s molecular interaction networks, highlighting key partners, regulatory feedback loops (such as AG–KNU–WUS), and the interplay between auxin and cytokinin signaling in developmental control.

Major Recommendations

To enhance structural flow and readability, the review would benefit from improved organization. The current mix of detailed mechanistic data with historical context disrupts the narrative. I suggest grouping content under clear thematic headings: (1) Structural Features, (2) CRC in Meristem Termination, (3) CRC in Auxin/Cytokinin Crosstalk, (4) Interaction Networks, (5) Evolutionary Diversification, (6) Polymorphisms and Natural Variation, and (7) Outlook.

Please consolidate repeated explanations, such as the AG–KNU–WUS feedback loop, into a single subsection to improve clarity. Conciseness can be increased by replacing overly detailed descriptions (e.g., exhaustive lists from Y2H assays) with tables or figures, such as a summary table of CRC-interacting proteins categorized by species, method, and function.

While species-specific comparisons are well described, they could be made more accessible through visual aids, such as comparative boxes or schematics illustrating direct versus indirect WUS repression, the presence or absence of nectary regulation, and conserved versus novel CRC functions. Sections that read like primary research (e.g., step-by-step experimental results) should be condensed into mechanistic conclusions and biological implications to maintain the review's style.

Finally, adding a conceptual diagram of the CRC regulatory network that integrates AG, KNU, TRN2, WUS, auxin, cytokinin, and other partners, along with an evolutionary tree mapping CRC/DL functions across species, would significantly enhance visual comprehension.

Minor Recommendations

Always define abbreviations at their first use in each central section (e.g., FM, CZ, CRM). Please ensure that Figure 1 is placed correctly in the middle and is not cropped on the external right. Lastly, verify that all in-text citations correspond to the correct and most recent sources, especially in instances where numbers cluster (e.g., [25,26] vs. [26,79]).

Author Response

Response to reviewer comments:

All Authors are very grateful for constructive comments provided by Reviewer. Presented suggestions are very valuable and are analysed carefully to improve the manuscript. Following changes were introduced.

Rev 1

Major Recommendations

To enhance structural flow and readability, the review would benefit from improved organization. The current mix of detailed mechanistic data with historical context disrupts the narrative. I suggest grouping content under clear thematic headings: (1) Structural Features, (2) CRC in MeristemTermination, (3) CRC in Auxin/Cytokinin Crosstalk, (4) Interaction Networks, (5) EvolutionaryDiversification, (6) Polymorphisms and Natural Variation, and (7) Outlook.

Answer: Clear thematic heading were introduced (in blue color) as suggested. In red are marked previous headings to be removed.

Please consolidate repeated explanations, such as the AG–KNU–WUS feedback loop, into a single subsection to improve clarity. Conciseness can be increased by replacing overly detailed descriptions (e.g., exhaustivelists from Y2H assays) with tables or figures, such as a summary table of CRC-interacting proteins categorized by species, method, and function.

Answer: Repeated explanations are consolidated. In red are marked removed text fragments, while in blue is present the added text.  Figures nr 2, 3, 4, and 5 are added to address the complex regulatory mechanisms concerning AG–KNU–WUS and CRC activity. Also the Fig. 7 was added to show putative functional partners of CRC. The Figure 1 presents the domian organization of CRC. Moreover following Supplementary Materials were added:

Fig. S1. Multiple sequence alignment of 100 CRC protein sequences to show the strongest conservation of amino acid residues.

Table S1. Representative genes that are either directly regulated by AG or integrated within the AG regulatory network are listed.

Table S2. Genes and proteins associated with CRC regulation and the biological outcomes of their interactions.

File S1. Detailed information concerning CRC functional partners from Fig. 7.

While species-specific comparisons are well described, they could be more accessible through visual aids, such as comparative boxes or schematics illustrating direct versus indirect WUS repression, the presence or absence of nectary regulation, and conserved versus novel CRC functions. Sections that read like primary research (e.g., step-by-step experimental results) should be condensed into mechanistic conclusions and biological implications to maintain the review's style.

Answer: The sections were shortened to make the manuscript build more on mechanistic conclusions and biological implications. As in the point above, the removed text fragments are marked in red, while in blue is present the added text.  Figures nr 2, 3, 4, 5 and 7 are added to show complex mechanisms of CRC functions and WUS repression. Also  Table S2 address these issues..

Finally, adding a conceptual diagram of the CRC regulatory network that integrates AG, KNU, TRN2, WUS, auxin, cytokinin, and other partners, along with an evolutionary tree mapping CRC/DL functions cross species, could significantly enhance visual comprehension.

Answer: An evolutionary tree mapping CRC/DL functions across species was added as Fig. 8. Conceptual diagrams of the CRC regulatory Network, that integrates AG, KNU, TRN2, WUS, auxin, cytokinin, and other partners is presented on Fig. 3, 4, and 5.

The additional figures: nr 1 ( CRC domain organization), S1- multiple CRC aa sequence alignments and 7 CRC interaction Network containing functionally related proteins were added, to improve the visual comprehension

Minor Recommendations

Always define abbreviations at their first use in each central section (e.g., FM, CZ, CRM). Please ensure that Figure 1 is placed correctly in the middle and is not cropped on the external right. Lastly, verify that all in-text citations correspond to the correct and most recent sources, especially in instances where numbers cluster (e.g., [25,26] vs. [26,79]).

Answer: Abbreviations were defined at the first use. The Fig. 1 is placed in the middle and is not on the external right. Citations correspond to the correct sources.

Reviewer 2 Report

Comments and Suggestions for Authors

The manuscript would be more attractive if it contained more figures, schemes, etc. The classical scheme ‘Introduction, Results’ is not quite good for Review since these two sections here very logically similar. The comments are listed below. They are marked by the line or line range.

13-14

… CRABS CLAW (CRC) is a member of the plant-specific YABBY transcription factor family, defined by the presence of a C2C2 zinc-finger domain and a C-terminal YABBY domain.

First, I suspect that these are DNA binding domains. Second, the YABBY family is plant-specific and belongs to the fourth superclass Other all-alpha-helical DNA binding domains, the class High-mobility group (HMG) domain factors.  This class is common between mammals and plants.

Blanc-Mathieu, R., Dumas, R., Turchi, L., Lucas, J., & Parcy, F. (2024). Plant-TFClass: a structural classification for plant transcription factors. Trends in plant science, 29(1), 40–51. https://doi.org/10.1016/j.tplants.2023.06.023

138-141

…The functional activity of transcription factors is frequently mediated not by monomeric forms but through the formation of homo- or heterodimers, which may assemble in the cytoplasm prior to nuclear translocation or upon binding to closely spaced cis-regulatory motifs within evolutionarily conserved promoter regions [52,53].

->

The functional activity of transcription factors is frequently mediated not by monomeric forms but through the formation of homo- or heterodimers, which may assemble in the cytoplasm prior to nuclear translocation or upon binding to closely spaced cis-regulatory motifs within conserved promoter regions [52,53].

I suspect that action of transcription factors (TFs), as well as their homo- or heterodimers is not restricted to the evolutionarily conserved promoter regions, promoters often are not evolutionarily conserved.

95, 167

The text concerning the ref. [35] is duplicated. You twice mention in the Introduction and Results the same information on 48 TFs among the total 140 proteins. Try to be succinct. Also, the standard format “Introduction, Results” is not good for your paper which is review, at least you should left an extended abstract in Introduction and all major content in Results

181

Among the proteins identified as binding to the CRC promoter in yeast

->

Among the proteins identified as binding to the CRC promoter in yeast Y1H screening – (I suspect so, correct this)

235

Yeast two-hybrid (Y2H) screens… But the 1st occurrence of Y2H is in the line 218.

359

The regulation of YUC4 by CRC involves its binding to YABBY-binding motifs (GA[A/G]AGAAA)

Here you need the ref. like this

Shamimuzzaman, M., & Vodkin, L. (2013). Genome-wide identification of binding sites for NAC and YABBY transcription factors and co-regulated genes during soybean seedling development by ChIP-Seq and RNA-Seq. BMC genomics, 14, 477. https://doi.org/10.1186/1471-2164-14-477

409

PDF, Fig.1 is out of margin in a page

559

The manuscript has only one Figure, I propose make all text more graphically friendly, e.g. subsection 2.6. Phylogenetic Insights into CRC Gene Lineage ask for a phylogenetic tree, an alignment or something like them.

679 misprint

  1. Orashakova

Author Response

Rev. 2

Authors are very grateful for constructive and valuable  comments provided by Reviewer. Presented suggestions were analysed carefully to improve the manuscript. Following changes were introduced.

The manuscript would be more attractive if it contained more figures, schemes, etc. The classical scheme ‘Introduction, Results’ is not quite good for Review Since these two sections here are very logically similar. The comments are listed below. They are marked by the line or linerange.

Answer: The classical scheme was changed. The removed text is marked in red, the added text is marked in blue.

13-14

… CRABS CLAW (CRC) is a member of the plant-specific YABBY transcription factor family, defined by the presence of a C2C2 zinc-finger domain and a C-terminal YABBY domain.

First, I suspect that these are DNA Binding domains. Second, the YABBY family is plant-specific and belongs to the fourth superclass Other all-alpha-helical DNA Winding domains, the class High-mobilitygroup (HMG) domain factors.  This class is common between mammals and plants.

Blanc-Mathieu, R., Dumas, R., Turchi, L., Lucas, J., &Parcy, F. (2024). Plant-TFClass: a structuralclassification for plant transcriptionfactors. Trends in plant science, 29(1), 40–51. https://doi.org/10.1016/j.tplants.2023.06.023

Answer: Following fragment and suggested citation were added in section 2. (marked in blue).

The YABBY family is plant-specific and belongs to the fourth superclass; Other all-alpha-helical DNA bindingdomains, the class High-mobility group (HMG) domain factors. This class is common between mammals and plants [27].

138-141

…The functional activity of transcription factors is frequently mediated not by monomeric forms but through the formation of homo- or heterodimers, chich may assemble in the cytoplasm prior to nuclear translocation or upon binding to closely spaced cis-regulatory motifs with in evolutionarily conserved promoter regions [52,53].

->

I suspect that action of transcription factors (TFs), as well as their homo- or heterodimers is not restricted to the evolutionarily conserved promoter regions, promoters often are not evolutionarily conserved.

Thank you  for suggestion. Words „evolutionary conserved” were removed. This is now on page nr 14.

95, 167

The text concerning the ref. [35] is duplicated. You twice mention in the Introduction and Results the same information on 48 TFs among the total 140 proteins. Try to be succinct. Also, the standard format “Introduction, Results” is not good for your paper which is review, at least you should left an extender abstract in Introduction and all major content in Results

Thank you  for comments. The duplicated fragment was removed from the Introduction. The removed fragment was  marked in red. Abstract is strongly shortened (removed fragments in red), heading „Results” is removed, instead of heading Discussion the Outlook was applied..

181

Among the proteins identified as binding to the CRC promoter in yeast

->

Among the proteins identified as binding to the CRC promoter in yeast Y1H screening – (I suspectso, correctthis)

Thank you for suggestion. This fragment was removed. The informations related to proteins interacting with CRC and its promoter are presented in Table S2.

235

Yeast two-hybrid (Y2H) screens… But the 1st occurrence of Y2H is in the line 218.

Answer: The first occurrence of Yeast two-hybrid is now on page 14, lower part. After this place the abbreviation Y2H is used.

359

The regulation of YUC4 by CRC involves its binding to YABBY-binding motifs (GA[A/G]AGAAA)

Here you need the ref. Like this

Shamimuzzaman, M., &Vodkin, L. (2013). Genome-wideidentification of bindingsites for NAC and YABBY transcriptionfactors and co-regulatedgenesduringsoybeanseedling development by ChIP-Seq and RNA-Seq. BMC genomics, 14, 477. https://doi.org/10.1186/1471-2164-14-477

Thank you for the comment. Reference was added as nr 104, now on page nr 12.

409

PDF, Fig.1 is out of margin in a page

Fig nr 1 in slightly reduced and is within margins.

Answer- localizartion of Fig. 1 and other Figures is corrected.

559

The manuscript Has only one Figure, I propose make all text more graphically friendly, e.g. subsection 2.6. Phylogenetic insights into CRC Gene Lineage ask for a phylogenetic tree, an alignment or some Thing like them.

Answer: Several Figures (nr 1,2,3,4,5,7,8 and S1) were added. Tables S1 and S2 were added. They are now included into the revised manuscript. The additional figures, nr 1( CRC domain organization), S1- multiple CRC aa sequence alignments and 7 presenting CRC functional interaction Network containing functionally related proteins were provided. Also Fig 2-5 were added to show complex mechanisms of WUS, AG and CRC action. Moreover,  Figure nr 8 shows evolutionary history of CRC/DL genes and Tables S1 and S2 show genetic and protein interactions of AG and CRC.

679 misprint

  1. Orashakova

Answer: Misprint corrected.

Round 2

Reviewer 1 Report

Comments and Suggestions for Authors

This revision has been considerably improved and can be published.